# Whole genome sequencing of SARS-CoV2 strains circulating in Iran during five waves of pandemic

Jila Yavarian[1], Ahmad Nejati[1], Vahid Salimi[1], Nazanin Zahra Shafiei Jandaghi[1], Kaveh Sadeghi[1], Adel Abedi[2], Ali Sharifi Zarchi[3], Mohammad Mehdi Gouya[4], Talat Mokhtari-Azad📷[1]*

1 Virology Department, School of Public Health, Tehran University of Medical Sciences, Tehran, Iran, 2 Mathematics Department, Shahid Beheshti University, Tehran, Iran, 3 Department of Computer Engineering, Sharif University of Technology, Tehran, Iran, 4 Iranian Center for Communicable Disease Control, Tehran, Iran

* mokhtari@tums.ac.ir

## Abstract

### Purpose

Whole genome sequencing of SARS-CoV2 is important to find useful information about the viral lineages, variants of interests and variants of concern. As there are not enough data about the circulating SARS-CoV2 variants in Iran, we sequenced 54 SARS-CoV2 genomes during the 5 waves of pandemic in Iran.

### Methods

After viral RNA extraction from clinical samples collected during the COVID-19 pandemic, next generation sequencing was performed using the Nextseq platform. The sequencing data were analyzed and compared with reference sequences.

### Results

During the 1st wave, V and L clades were detected. The second wave was recognized by G, GH and GR clades. Circulating clades during the 3rd wave were GH and GR. In the fourth wave GRY (alpha variant), GK (delta variant) and one GH clade (beta variant) were detected. All viruses in the fifth wave were in clade GK (delta variant). There were different mutations in all parts of the genomes but Spike-D614G, NSP12-P323L, N-R203K and N-G204R were the most frequent mutants in these studied viruses.

### Conclusions

These findings display the significance of SARS-CoV2 monitoring to help on time detection of possible variants for pandemic control and vaccination plans.

**Data Availability Statement:** All submitted files are available from the GISAID database (Accession numbers: EPI-ISL-1014676-87, EPI-ISL-959275-84, EPI-ISL-862075-81, EPI-ISL-1993547-557,

EPI-ISL-2360250-57, EPI-ISL-4803556-58, EPI-ISL-4803554, EPI-ISL-4803538, and EPI-ISL-4803528).

**Funding:** A part of this study is supported by National Institute for Medical Research Development (NIMAD) under grant number 994376. The funders had no role in study design, data collection and analysis, decision to publish, or preparation of the manuscript.

**Competing interests:** The authors have declared that no competing interests exist.

# 1 Introduction

In December 2019, severe acute respiratory syndrome coronavirus 2 (SARS-CoV2) was detected in Wuhan, China which has known as the cause of coronavirus disease 2019 (COVID-19) pandemic [https://covid19.who.int/].

To date, Iran has experienced five waves of the pandemic since the first detection of SARS-CoV2 on 19 February 2020 in Iran [1]. As of 16 November 2021, 253,640,693 cases of SARS-CoV2 with 5,104,899 deaths were reported worldwide and in Iran 6,045,212 laboratory confirmed cases with 128,272 deaths reported [https://covid19.who.int/].

SARS-CoV2 is a virus in *coronaviridae* family with a single stranded RNA. The genome is about 30 kb and consists of genes encoding multiple non-structural, structural and accessory proteins. The non-structural proteins include NSP1-16 which are necessary for virus transcription and replication. These proteins are encoded by ORF1ab with about 21300 nucleotides length [2].

NSP1 helps virus to evade innate host antiviral response and promote viral growth. NSP2 can bind to two host proteins prohibitin 1 and prohibitin 2 that are recognized to be important in cell migration, cell cycle progression, apoptosis, cellular differentiation, and mitochondrial biogenesis [3]. NSP3 is a papain-like protease and also it can suppress host protein synthesis. NSP3 and NSP4 with other cofactors are important for virus replication by prompting membrane rearrangement. NSP5 is a cysteine-like protease which is the virus main protease and cleaves NSP4-NSP16. NSP6 is involved in autophagy. NSP7 and NSP8 form complex with NSP12 for viral replicase machinery. NSP7 has primer-independent RNA polymerase activity. NSP8 has primase activity- Primase is an enzyme that synthesizes primers during replication. NSP9, an RNA-binding protein, in complex with NSP8, is involved in RNA replication and virulence. NSP10 is a cofactor for the 2′ O-methyltransferase activity of NSP16 that increases evasion of the innate immune system, and the N7-guaninemethyltransferase/exoribonuclease activities of NSP14. NSP11 and NSP15 are involved in endoribonuclease activity and essential for replication. NSP12 has RNA polymerase activity. NSP13 has helicase and NTPase activity [2].

The structural proteins include spike (S), envelope (E), membrane (M) and nucleoprotein (N). Spike with a length of 1273 amino acid (aa), mediates attachment and entry. Envelope (75 aa) is a small membrane protein and important in virus infectivity. Membrane (222 aa) is important in virion morphogenesis and nucleoprotein (419 aa) is a viral genome packaging protein [2].

There are nine accessory proteins encoded by ORF3a, 3b, 6, 7a, 7b, 8, 9a, 9b and 10, which seem to be important in virus pathogenesis [2].

During virus replication, mutations can occur which might lead to alteration in protein functions and virus transmission and pathogenesis. Nowadays next-generation sequencing (NGS) is an effective method to identify different mutations and new variants for epidemiological and surveillance studies. SARS-CoV2 has different variants known as variants of concern (VOC), variants of interests (VOI) and variants under monitoring (VUM). It has also classified to different clades that currently 9 clades have been recognized based on markers mutations of the genome: S, L, V, G, and later of G into GH, GR and GV, and more recently GR into GRY and GRA [4]. Different variants of the SARS-CoV2 have been identified during the pandemic. Some spread worldwide, while others quickly faded away. Identification of the circulation of variants in a society is important, therefore, we set up NGS for SARS-CoV2 detected during five waves of pandemic for better understanding of circulation of different variants, genetic diversity and mutations in all non-structural, structural and accessory genes.

## 2 Materials and methods

### 2.1 Sample selection

Throat swab specimens from COVID-19 suspected patients were sent for sequencing to National Influenza Center (NIC) located at Virology Department, School of Public Health, Tehran University of Medical Sciences, Tehran, Iran. After primary detection by Real time PCR the samples from different parts of the country with ct value lower than 25 were selected for NGS. Iran experienced the 1st wave of SARS-CoV2 pandemic in February until May 2020. The 2nd wave began in late June until September 2020. The 3rd wave occurred during October to December 2020. The 4th wave started on early April until June 2021 and the 5th wave was from August until October 2021. This study was approved by ethics committee of National Institute for Medical Research Development, Tehran, Iran (IR.NIMAD.REC.1399.119).

### 2.2 Next generation sequencing (NGS)

After viral RNA extraction using High Pure Viral Nucleic Acid kit (Roche,Germany) according to the manufacturer's instruction, cDNA synthesis was performed. Library construction was done by using the Nextera DNA Flex kit (Illumina,USA), then used for hybridization using Respiratory Virus Oligo Panel kit (Illumina,USA). This was followed by bead-based capture of hybridized probes, amplification and clean-up. After quality control assessment for library concentration by Qubit (Thermo Fisher,USA), sequencing was performed using Next-Seq 550 machine (Illumina,USA).

### 2.3 Data & phylogenetic analysis

For data analysis, all collection of reads were mapped against the reference genome assembly of SARS-CoV2. We had high quality of assembled full viral genome coverage without undetermined nucleotides. The assembled genomes were analyzed using CoVsurver mutations App in GISAID [4] and aligned by BioEdit sequence alignment software. After sequence alignment, a phylogenetic tree was drawn by neighbor joining method (bootstrap 1000) using MEGA v7 software.

Then all 54 Iranian full genome SARS-CoV2 of different waves were submitted to GISAID with accession numbers: EPI-ISL-1014676-87, EPI-ISL-959275-84, EPI-ISL-862075-81, EPI-ISL-1993547-557, EPI-ISL-2360250-57, EPI-ISL-4803556-58, EPI-ISL-4803554, EPI-ISL-4803538 and EPI-ISL-4803528.

## 3 Results

Fifty four COVID-19 confirmed cases were subjected to NGS selecting 10, 10, 9, 20 and 5 samples from 1st, 2nd, 3rd, 4th and 5th waves respectively. Here we analyzed the mutations of each genes separately by comparing with hCoV-19/Wuhan/WIV04/2019 in GISAID.

### 3.1 Nonstructural proteins (ORF1ab)

The mutations of NSP genes are illustrated in the Fig 1. It is important to note that not all the mutations mentioned in the Fig 1 were found in all viruses during the related waves. Meanwhile in NSP10 and NSP11 no changes have been detected.

### 3.2 Structural proteins (S-E-M-N)

**Spike.** As shown in Table 1, amino acid substitutions in S glycoprotein were increased with pandemic progression. During the 1st wave there was no special amino acid substitutions

**Table 1. Amino acid changes detected in spike glycoprotein during the five waves of SARS-CoV2 pandemic in Iran.**

*Wave 1*

| Sample name | Amino acid changes in Spike | | |
|---|---|---|---|
| hCoV-19/Iran/Tehran-08/2020 | D614G | | |
| hCoV-19/Iran/Tehran-09/2020 | D614G | | |
| hCoV-19/Iran/Qom-907/2020 | D614G | | |
| hCoV-19/Iran/Gorgan-NIC992/2020 | D614G | | |
| hCoV-19/Iran/Gorgan-NIC118/2020 | D614G | | |
| hCoV-19/Iran/Gorgan-NIC028/2020 | D614G | G142S | I210del |
| hCoV-19/Iran/Gorgan-NIC140/2020 | D614G | | |
| hCoV-19/Iran/Hamedan-66H/2020 | D614G | | |
| hCoV-19/Iran/Urmia-715U/2020 | D614G | | |
| hCoV-19/Iran/Urmia-931U/2020 | D614G | G142S | I210del |

*Wave 2*

| Sample name | Amino acid changes in Spike | | | | | | |
|---|---|---|---|---|---|---|---|
| hCoV-19/Iran/Gilan-NIC189/2020 | D614G | | | | | | |
| hCoV-19/Iran/Ghilan-S15-319/2020 | D614G | | | | | | |
| hCoV-19/Iran/Ghilan-T15-191/2020 | D614G | | | | | | |
| hCoV-19/Iran/Tehran-055M/2020 | D614G | | | | F1256L | I210del | L5F |
| hCoV-19/Iran/Gilan-NIC184/2020 | D614G | | | | | | |
| hCoV-19/Iran/Gilan-NIC230/2020 | D614G | | | | | I210del | |
| hCoV-19/Iran/Ghilan-M30-241/2020 | D614G | | | | | I210del | |
| hCoV-19/Iran/Gilan-NICS1-58/2020 | D614G | D574N | D950N | Q677H | | | |
| hCoV-19/Iran/Ghilan-S1-32/2020 | D614G | | | | | | |
| hCoV-19/Iran/Qom-629/2020 | D614G | | | | | I210del | |

*Wave 3*

| Sample name | Amino acid changes in Spike | | | | | | | |
|---|---|---|---|---|---|---|---|---|
| hCoV-19/Iran/Tehran-NIC45RC/2020 | D614G | | I210del | | | | | |
| hCoV-19/Iran/Tehran-NIC12RC/2020 | D614G | | | A845S | | | | |
| hCoV-19/Iran/Tehran-04/2020 | D614G | Q314R | I210del | | | | | |
| hCoV-19/Iran/Tehran-06/2020 | D614G | | | | A846V | D138Y | M177I | S477N |
| hCoV-19/Iran/Tehran-01/2020 | D614G | | | | | D138Y | | S477N |
| hCoV-19/Iran/Tehran-02/2020 | D614G | | | | | | | |
| hCoV-19/Iran/Tehran-03/2020 | D614G | | I210del | | | | | |
| hCoV-19/Iran/Tehran-05V/2020 | D614G | | | | | D138Y | | S477N |
| hCoV-19/Iran/Kashmar-15K/2020 | D614G | | I210del | | | | | |

*Wave 4*

| Sample name | Amino acid changes in Spike | | | | | | | | | | | |
|---|---|---|---|---|---|---|---|---|---|---|---|---|
| hCoV-19/Iran/Tehran-NIC-K23/2021 | D614G | A570D | D1118H | H69del | N501Y | P681H | S982A | T716I | V70del | Y144del | | |
| hCoV-19/Iran/Hormozghan-NIC-15/2021 | D614G | A570D | D1118H | H69del | N501Y | P681H | S982A | T716I | V70del | Y144del | | |
| hCoV-19/Iran/Tehran-NIC-24/2021 | D614G | A570D | D1118H | H69del | N501Y | P681H | S982A | T716I | V70del | Y144del | I100T | L699I |
| hCoV-19/Iran/Tehran-NIC-V30/2021 | D614G | A570D | D1118H | H69del | N501Y | P681H | S982A | T716I | V70del | Y144del | | |
| hCoV-19/Iran/Tehran-NIC-SH17/2021 | D614G | A570D | D1118H | H69del | N501Y | P681H | S982A | T716I | V70del | Y144del | | |
| hCoV-19/Iran/Tehran-NIC-30/2021 | D614G | A570D | D1118H | H69del | N501Y | P681H | S982A | T716I | V70del | Y144del | | L699I |
| hCoV-19/Iran/Tehran-NIC-23K/2021 | D614G | A570D | D1118H | H69del | N501Y | P681H | S982A | T716I | V70del | Y144del | | |

*(Continued)*

**Table 1.** (Continued)

| Sample name | Amino acid changes in Spike |
|---|---|
| hCoV-19/Iran/Tehran-NIC-28/2021 | D614G, A570D, D1118H, H69del, N501Y, P681H, S982A, T716I, V70del, Y144del, I100T, L699I |
| hCoV-19/Iran/Tehran-NIC-31/2021 | D614G, A570D, D1118H, H69del, N501Y, P681H, S982A, T716I, V70del, Y144del |
| hCoV-19/Iran/Tehran-NIC-17/2021 | D614G, A570D, D1118H, H69del, N501Y, P681H, S982A, T716I, V70del, Y144del, I100T, L699I |
| hCoV-19/Iran/Tehran-NIC-15/2021 | D614G, A570D, D1118H, H69del, N501Y, P681H, S982A, T716I, V70del, Y144del, I100T, L699I |
| hCoV-19/Iran/Kerman-NIC-K1/2021 | D614G, A570D, D1118H, H69del, N501Y, P681H, S982A, T716I, V70del, Y144del |
| hCoV-19/Iran/Kerman-NIC-K2/2021 | D614G, A570D, D1118H, H69del, N501Y, P681H, S982A, T716I, V70del, Y144del, I100T, L699I |
| hCoV-19/Iran/Tehran-NIC-S7/2021 | D614G, A570D, D1118H, H69del, N501Y, P681H, S982A, T716I, V70del, Y144del, I100T, L699I, K1191N |
| hCoV-19/Iran/Hormozghan-NIC-B7/2021 | D614G, A243del, A701V, D80A, D215G, E484K, K417N, L242del, L244del, N501Y |
| hCoV-19/Iran/Boshehr-NIC-P10/2021 | D614G, D950N, E156G, F157del, G142D, L452R, P681R, R158del, T19R, T478K |
| hCoV-19/Iran/Boshehr-NIC-P11/2021 | D614G, D950N, E156G, F157del, G142D, L452R, P681R, R158del, T19R, T478K |
| hCoV-19/Iran/Yazd-NIC-Y3/2021 | D614G, D950N, E156G, F157del, G142D, L452R, P681R, R158del, T19R, T478K, E202Q, T95I, C1250R, D1257F, F1256L |
| hCoV-19/Iran/Maku-NIC-M5/2021 | D614G, D950N, E156G, F157del, G142D, L452R, P681R, R158del, T19R, T478K, T95I, S255F |
| hCoV-19/Iran/Yazd-NIC-Y6/2021 | D614G, D950N, E156G, F157del, G142D, L452R, P681R, R158del, T19R, T478K, E202Q, T95I, L938F |
| *Wave 5* | |
| hCoV-19/Iran/Maragheh-NIC-S2/2021 | D614G, D950N, P681R, T19R, T478K, T95I, I850L |
| hCoV-19/Iran/Ardakan-NIC-S5/2021 | D614G, D950N, E156G, F157del, G142D, L452R, P681R, R158del, T19R, T478K, T95I, D574Y, T299I |
| hCoV-19/Iran/North Khorasan-NIC-S6/2021 | D614G, D950N, E156G, F157del, G142D, L452R, P681R, R158del, T19R, T478K, T29A, T250I |
| hCoV-19/Iran/Ahvaz-NIC-S8/2021 | D614G, D950N, E156G, F157del, G142D, L452R, P681R, R158del, T19R, T478K, T95I, A262S, I850L |
| hCoV-19/Iran/Ahvaz-NIC-S9/2021 | D614G, D950N, E156G, F157del, G142D, L452R, P681R, R158del, T19R, T478K, T95I, A262S, I850L |

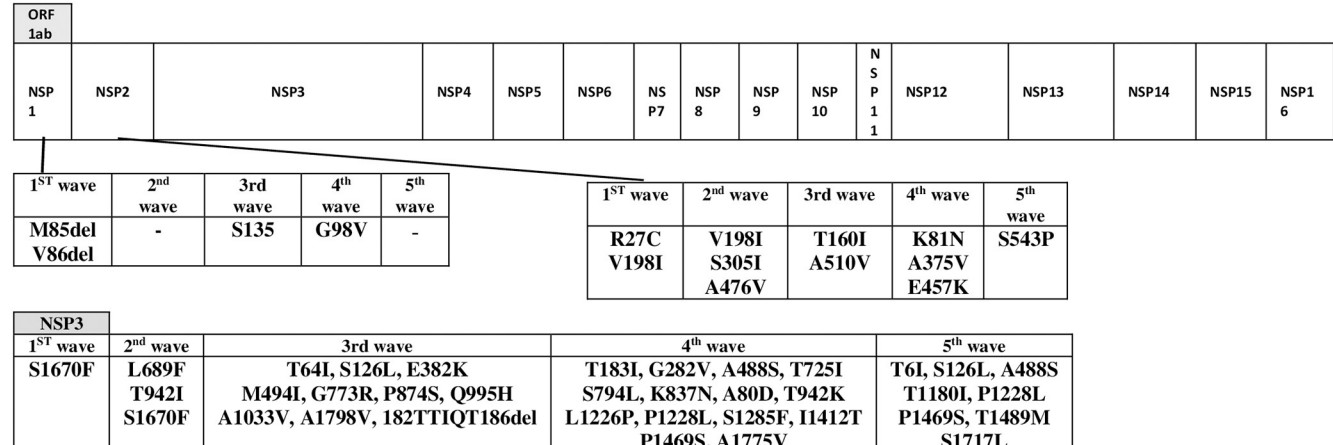

**Fig 1. The substitutions of nonstructural proteins during the five waves of SARS-CoV2 pandemic in Iran.**

on the S glycoprotein but viruses detected during the 4th and 5th waves had the most frequent amino acid substitutions. It should be noted that D614G was detected in all viruses after the 1st wave.

**Envelope.** No mutation was found in viruses studied during the 1st, 2nd, 3rd and the 5th waves but in two samples in the 4th wave there were mutations encoding the amino acid

substitution P71L in one sample belonged to beta variant and S68F in one sample belonged to alpha variant.

**Membrane.** During the 1st and the 2nd waves, there were no mutations in M, but a mutation causing the amino acid substitution I73M was detected in three samples during the 3rd wave. Four samples during the 4th wave and 4 samples of the 5th wave had I82T which all these 8 samples belonged to delta variant.

**Nucleoprotein.** During the 1st wave, one sample had N192K in N and two samples had A35V. During the 2nd wave, two samples had D3Y and M234I, three samples had S194L substitutions. One sample had P13T and S194L. Another sample had A220V and P326L. hCoV-19/Iran/Gilan-NICS1-58/2020 along with four viruses during the 3rd wave, had R203K and G204R. hCoV-19/Iran/Tehran-05V/2020 had P162S beside R203K and G204R. One sample during the 3rd wave, had S201I. Three samples had S194L along with another substitution including G34W, P383L and P13T. During the 4th wave, 13 samples of alpha variant had D3L, R203K, G204R and S235F. One sample of alpha variant beside these four substitutions had also L219F. Seven samples belonged to delta variant had D63G, D377Y, G215C and R203M. One sample of delta variant during the 5th wave had D63G, R203M and D377Y. One sample had just R203M and D377Y and one sample had D63G, R203M and G215C. The only beta variant had T205I and T362I.

## 3.3 Accessory proteins

**First wave.** One sample had T151I in NS3, another sample had A105V in NS7a and two samples had T40I in NS7b.

**Second wave.** In NS3, five samples had Q57H and one sample had G18V along with Q57H. One sample had G254V and one sample had T223A and V112F, therefore during the 2nd wave eight samples had just mutation in NS3 among all accessory proteins.

**Third wave.** In NS3, two samples had Q57H the same as five samples in the 2nd wave. Two more samples had one more substitution along with Q57H in NS3, including W131C and T223I. One sample had just N58G in NS3. Three samples had a deletion at G66 (G66del) and S67 (S67del) and K68E in NS8. One sample had T223I and L4F in NS3.

**Fourth wave.** During the fourth wave, amino acid substitutions detected in NS3 were W131C, S171L, G100C, D238Y, S26L, Q57H and S171L in different samples. There was just W27L in NS6 in one sample. T39I, T120I, V82A and S83L were detected in NS7a. NS7b had T40I and a stop codon at E39 (E39stop). NS8 had the most frequent amino acid substitutions in viruses circulating during the 4th wave including stop codons at K68 (K68stop) and Q27 (Q27stop) and R52I, V62L, Y73C, C90F. These stop codons at NS7b and NS8 may lead to producing the truncated proteins. During the 5th wave, amino acid substitutions were detected in NS3a, NS7a and NS7b which K16T, S26L and L41F in NS3a, L49P, A50D, D51H, T120I, V82A in NS7a and T40I in NS7b were found.

## 3.4 Phylogenetic analysis

Fig 2 shows the phylogenetic analysis of 54 SARS-CoV2 viruses circulating during the different waves in Iran. Analyses showed that during the 1st wave clades V and L were circulating which V was dominant. During the 2nd wave, GH was dominant, but G and GR were also detected. Surprisingly one V clade was also detected in the 2nd wave. GH and GR were the clades during the 3rd wave. During the 4th wave, 14 viruses of alpha variant were belonged to GRY, five delta variants were in GK clade and one beta variant was in GH clade and finally five delta variants of the 5th wave were in GK clade.

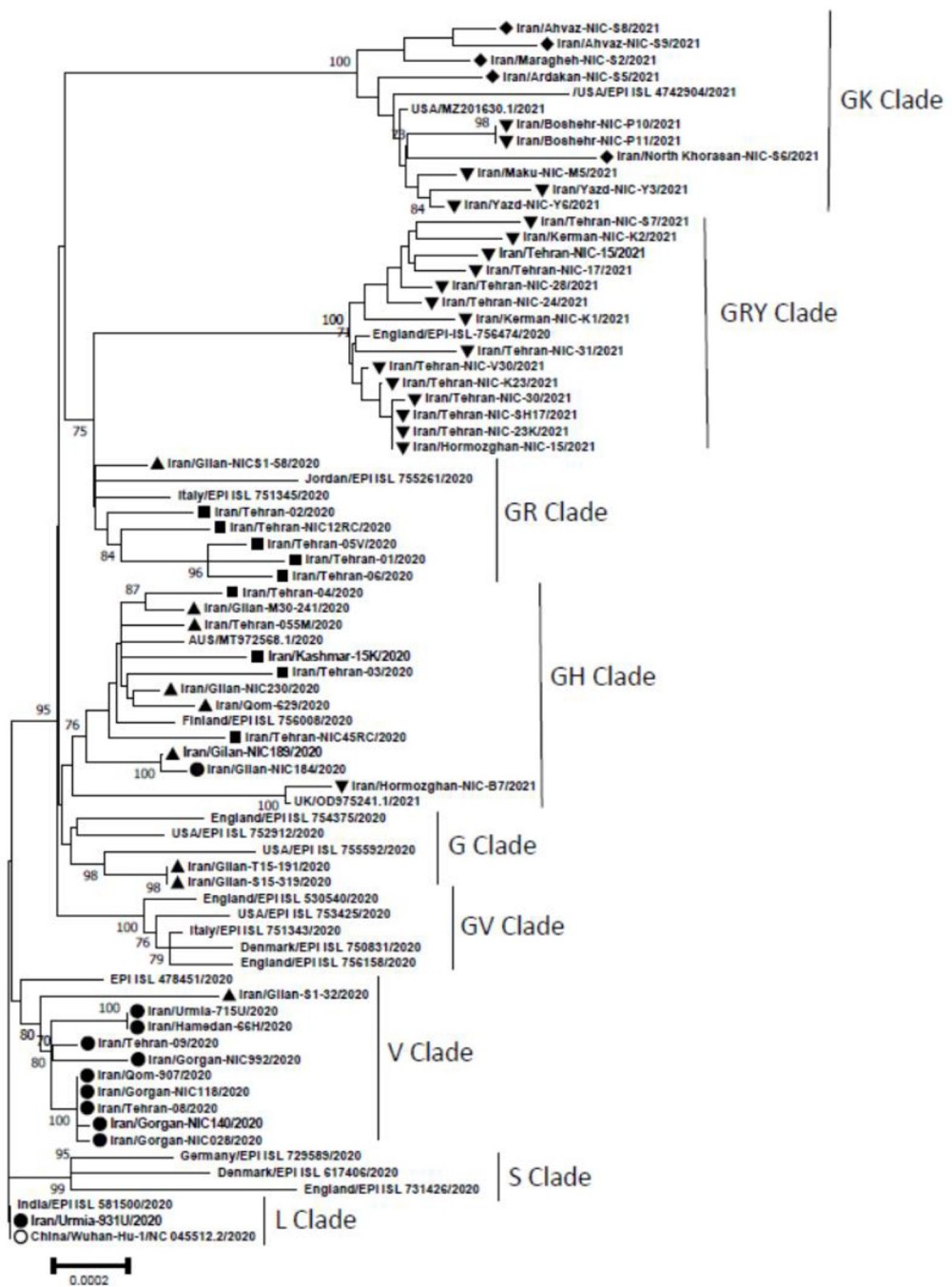

**Fig 2. Phylogenetic tree of SARS-CoV2 full-length genomes constructed by MEGA 7.** The Neighbor joining method was used with 1,000 bootstrap replicates. The tree contains 54 SARS-CoV2 sequences of these study compared to the reference sequence from GISAID and some other sequences from each clade. In this tree the reference sequence is marked by white circle and sequences of this study were marked as follow: The 1st wave black circle, the 2nd wave inverted black triangle, the 3rd wave with black square, the 4th wave with black triangle and the 5th wave with black diamond.

## 4 Discussion

In this study, we reported the circulation of distinct clades of SARS-CoV2 during the five waves in Iran. The V clade was found during the 1st wave. Clade V is characterized by NSP6-L37F plus NS3-G251V. At the end of the first wave, one sample detected in clade L (reference clade). The 2nd wave was recognized by G, GH and GR clades. G clade shows D614G in spike which is one of the most important substitutions in spike. Clade GH has S-D614G plus NS3-Q57H substitutions and clade GR characterized by S-D614G and N-G204R. One sample belonged to the V clade was also detected in the 2nd wave. Circulating clades during the 3rd wave were GH and GR. During the fourth wave, GH (1 beta), GRY (alpha) and GK (delta) clades were detected. GRY clade (alpha) has S-H69del, S-V70del, S-Y144del, S-N501Y plus S-D614G and N-G204R. All viruses in the 5th wave were in clade GK (S-D614G plus S-T478K).

In different clades NGS of 54 samples selected during the five waves in Iran showed important mutations in the different parts of the genome which all were analyzed and compared with circulating variants worldwide. The effects of important mutations were discussed here.

ORF1ab is more than two thirds of the SARS-CoV2 genome with 21,290 nucleotides at the 5' end which encodes 16 non-structural proteins (NSP1-NSP16). Among these NSPs, NSP3 had the highest number of mutations. NSP3 is important for virus replication and it can suppress host protein synthesis, then amino acid substitutions in NS3 deserve greater study in vitro.

Suppression of type I interferon (IFN) response is a consequence of infection of SARS-CoV2. NSP5, main protease, is an IFN antagonist. NSP5 variant K90R seen in SARS-CoV2 retained the IFN-antagonizing activity. The suppressive outcome of NSP5 on IFN-β gene transcription induced by IKKϵ, TBK1, RIG-I and MAVS suggested that NSP5 probably involved at a stage downstream of IRF3 phosphorylation in the cytoplasm [5]. In this study K90R was detected just in one sample during the 4th wave while it has been more often found in Icelandic and Chinese strains [6].

NSP6 is important for viral assembly, viral protein folding and replication. NSP6-L37F leads to asymptomatic transmission and reduced virulence which we can see in the most of the viruses detected during the 1st wave and one in the 2nd wave [7].

In hCoV-19/Iran/Gilan-NIC230/2020, E50G in NSP7 was detected in the 2nd wave which is important in immune response as shown in a study that amino acid residues 36–50 (HNDILLAKDTTEAFE) of NSP7 and also NSP13 and N are SARS-CoV2 specific T cell epitopes recognized by CD8 T cells [8].

NSP12 is necessary for the replication/transcription of the SARS-CoV2 genome, and this protein is considered as a target for the treatment of COVID-19. The P323L substitution of NSP12, could induce structural changes and adverse effect on proofreading during the replication of the virus. Meanwhile, the P323L is located in a pocket that might be the site of drug function [9]. In our research NSP12-P323L located in the NSP8 binding cleft [10] was detected after the first wave. NSP12-P323L is the most common detected substitution with increasing in occurrence over time [11].

A study by Mohammad., et al. showed that remdesivir has higher binding affinity to NSP12 with P323L than the wild-type RdRp, therefore they suggested using remdesivir in patients infected with SARS-CoV2 carrying P323L in NSP12 [12].

In this study there was no virus seen to contain F480, V557 and S861 in NSP12, which have shown to be important in decreasing susceptibility to remdesivir [11].

As all viruses studied in this research, did not have any mutations in the genes for NSP10 and NSP11, therefore these two nonstructural proteins could be considered as good targets for

diagnosis and targets for therapeutic agents. Furthermore, there were some mutations in other NSPs with that of unknown importance. The effect of these substitutions need to be studied in vitro.

After the 1st wave, all viruses studied in this research had S-D614G. Studies showed that the S-protein amino acid substitution, D614G, evolved concurrently with P323L in NSP12. NSP12- P323L might decrease the replication of viral RNA and therefore increase the probability of asymptomatic infections and/or change in incubation period. Meanwhile the infectivity-enhancing D614G substitution in S protein could compensate this reduction in virus replication and help virus transmission even by asymptomatic individuals [13]. Flores-Alanis., et al. showed that D614G substitution in S protein, P323L in NSP12, and R203K and G204R in N protein had a substantial association with the disease severity [9]. Another study showed that D614G in the S protein increased the infectivity of SARS-CoV2 and associated with lower RT-PCR cycle thresholds. It showed high viral load in the upper respiratory tract, but not enhanced disease severity. Also they showed that G614-bearing virions are not intrinsically more resistant to neutralization by convalescent sera [14]. G614 of S protein is responsible for making the virus 2.4 times more contagious with higher viral load [9, 15]. hCoV-19/Iran/Tehran-055M/2020 in the 2nd wave had D614G and L5F. One study showed that variants with D614G and L5F had increased infectivity [16].

Evaluation of the mutations which lead to amino acid substitutions, showed that Spike-D614G, NSP12-P323L, N-R203K and N-G204R were the most frequent amino acid substitutions in these studied viruses and also worldwide which we could consider that they may increase SARS-CoV2 transmissibility as the pandemic progressed.

The S protein is crucial factor for the entry of SARS-CoV2 to the host cell which interacts with the angiotensin-converting enzyme 2 (ACE-2) receptor through its receptor binding domain (RBD) [17]. The S477N is the part of an epitope recognized by human neutralizing antibodies and located in the RBD. A study showed that S477N increases the affinity for the ACE-2 receptor [9, 18]. Singh., et al. showed that S477N strengthen the binding of SARS-COV2 spike to the ACE-2 receptor [19]. In this study, two samples had S477N which both belonged to GR clade.

Among the variations in the alpha variant, S-N501Y, is in the receptor binding site which was shown to increase the binding of SARS-CoV2 to the ACE-2 host receptor, leading to increased viral fitness and transmission [20–22].

One study showed that N439K, N501Y and S477N significantly reduced the neutralization activity of some monoclonal antibodies [23]. Ostrov showed that B.1.1.7 with N501Y had increased affinity for ACE-2 and A570D, D614G and S982A substitutions might increase virus fusion by decreasing the intermolecular stability of S1 and S2 [24].

The delta variant (GK clade) has ten amino acid substitutions, T19R, G142D, 156del, 157del, R158G, L452R, T478K, D614G, P681R and D950N in the spike protein [25]. T19R removes an N-glycosylation site at position 17 that might also affect antigenic properties. A study by Li., et al. showed that variants with L452R were resistant to some neutralizing antibodies [16]. Tchesnokova., et al. showed that L452R might result in stronger binding to the ACE-2 and escape from neutralizing antibodies [26].

Overall L452R reduces sensitivity to neutralizing antibodies, increases viral infectivity, transmissibility, spike stability, ACE-2 binding affinity and viral fusogenicity, therefore it supports viral replication [27–30].

Delta variant circulation was started at the end of the 4th wave and all viruses detected during the 5th wave in Iran were the delta variant.

The beta variant (GH clade) includes L18F, D80A, D215G, R246I, K417N, E484K, N501Y, D614G, and A701V in the spike protein, which K417N, E484K, and N501Y are in the RBD

and increase the binding affinity for the ACE-2 receptor and eventually the infectivity [31]. K417N and E484K have also essential role in viral escape from neutralizing antibodies [32]. Overall K417N, E484K and L452R are vaccine escape mutants [33]. The only beta variant of this study did not have L18F and R246I but it had a deletion of three amino acids 242–244 (L242del, A243del and L244del) in spike. L242del, A243del and L244del showed reduced sensitivity to some neutralizing antibodies [22]. The beta variant is reported to have an increased risk of transmission and reduced neutralization by monoclonal antibody therapy, convalescent sera, and post-vaccination sera [34].

Some studies showed that most serum samples from vaccinated people or patients recovering from COVID-19 have shown full or slightly decreased capacity to inactivate SARS-CoV2 variants, except for variants with N501Y, K417N and E484K substitutions [22, 35–37]. We had just one beta variant during the 4th wave which had E484K in the S protein. As seen in different studies, E484K plays a crucial role in increasing virus transmission and decreasing antibody neutralizing titers [37–39], thus continuous screening for emerging variants with substitutions such as E484K is necessary for public health.

hCoV-19/Iran/Tehran-03/2020 in the 3rd wave had Q677H. Grabowski., et al. showed that E484K, F490S, S494P (in the RBD of spike) and Q677H and Q675H (in the vicinity of the polybasic cleavage site at the S1/S2 border) may limit efficiency of vaccines [40].

Liu., et al. showed that substitutions at residues T345, R346, K444, G446, N450, L452, S477, T478, E484, F486, and P499 were each related to the resistance to more than one monoclonal antibody, of which substitutions at S477 and E484 residues showed wide resistance [41].

Viruses with H69del and V70del had 2-fold higher infectivity compared to wildtype and also showed reduced neutralization sensitivity to mAb, targeting an as yet undefined epitope outside the RBD [42]. Sixteen samples of this study during the 4th wave had H69del, V70del and also Y144del. In one study it was shown that viruses with Y144del in spike had decreased sensitivity to convalescent sera [16].

Li., et al. showed that, among spike mutations, the most characteristic ones are substitutions such as D614G, N501Y, Y453F, N439K/R, P681H, K417N/T, and E484K, and deletions of ΔH69/V70 and Δ242–244, which enhanced viral infectivity, transmissibility, and resistance to neutralization [43]. Some of these mutations were found after the 2nd wave of pandemic in Iran.

P681R in the furin cleavage site may help in increased rate of membrane fusion, internalization and so better transmissibility which was found in delta variants in this study and worldwide [44]. Voss., et al. found that P681H of spike and S235F of nucleoprotein in the alpha variant changed the specificity of the corresponding epitopes [45]. During the 4th wave, 14 samples had P681H and S235F.

Mutations in the E gene were somewhat uncommon as in this study just two mutations in 2 different samples in the 4th wave were detected which needs more investigation to find out their effects on virus life cycle.

All viruses of delta variant in this study had I82T in M gene. Shen., et al. showed that this M gene substitution was more naturally fit, probably connected to glucose uptake during virus replication, and it is better to be included in genomic surveillance [25].

Nucleoprotein was mainly expressed in the initial stages of infection, and is important in viral RNA transcription and replication. Nucleoprotein has been shown to affect some basic cellular processes, inflammatory responses to upregulate the expression of the proinflammatory factor COX2, and it inhibits the innate immune responses in the host cell [46]. Therefore, amino acid substitutions in the nucleoprotein might have significant effect in immune response. R203K and G204R were the most common substitutions in the nucleoprotein in

viruses studied in this research and worldwide. These mutations are important in disease severity [47].

There are some studies about the accessory gene's mutations and their impact on the virus cell cycle. The results of research by Wu et al., showed that Q57H and G251V in NS3a, S194L and R203K/G204R in N made changes in the structure of proteins and also had effect on the binding affinity of intraviral protein-protein interactions during assembly and release of coronavirus. So these changes might be associated with virus evolution and beneficial for the virus and its virulence [48]. Some viruses during the waves 2,3 and 4 had NS3a-Q57H. In primary human respiratory cells, viruses with NS3a-Q57H evade stimulation of chemokine, cytokine, and interferon-stimulated gene expression [49]. Some studies have described that NS3a can competently induce apoptosis in the cell and affect virus uptake and release [50, 51]. Besides ORF3, some mutations were detected in different accessory genes of viruses in this study which more research is needed to explore their importance in virus pathogenesis. For instance, NS8 supposed to suppress immune responses [52], then substitutions of this protein can be imperative. Of special importance, stop codon mutations which lead to absence of NS8 can extend the duration of symptoms, then increase the virus transmissibility. Spike mutations such as D614G, HV69-70 del and L5F which affect the receptor binding affinity of the S protein and increase the virus transmission when associated with NS8 stop codons which extend the period of signs and symptoms might enhance the chance of transmission [53]. The combination of these variations were found especially in some viruses detected during the 4[th] wave which should raise concern.

In this study we did not have data on relationship between different SARS-CoV2 lineages and patient's signs and symptoms and disease severity which was an important limitation of this research.

In conclusion, we detected different lineages of SARS-CoV2 contributing to all five waves and showed that all viruses circulating during the 5[th] wave belonged to delta variant. We compared the mutations identified in our complete genomes study with those reported in GISAID. The findings of this study showed that with progression of the pandemic, the number of mutations were considerably increased which showed the adaptive evolution of SARS-CoV2 in human to increase transmissibility. Therefore genomic surveillance is an important tool to screen the progression of the COVID-19 pandemic. It should be noted that for variant detection, we partially sequenced S glycoprotein of more than 1000 samples with Sanger sequencer of which the results were compatible with NGS results during all 5 waves (unpublished data).

## Acknowledgments

Authors would like to express their great thanks to all data contributors, i.e. the authors and their originating laboratories responsible for obtaining the specimens, and their submitting laboratories for generating the genetic sequence and metadata and sharing via the GISAID Initiative, on which this research is based. We thank GISAID for all their support.

## Author Contributions

**Data curation:** Adel Abedi.

**Formal analysis:** Ali Sharifi Zarchi.

**Methodology:** Ahmad Nejati, Vahid Salimi, Nazanin Zahra Shafiei Jandaghi, Kaveh Sadeghi.

**Supervision:** Mohammad Mehdi Gouya, Talat Mokhtari-Azad.

**Writing – original draft:** Jila Yavarian.

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
