## [Decision Letter · Decision Letter 0]

20 Jan 2022

PONE-D-21-38222Whole genome sequencing of SARS-CoV2 strains circulating in Iran during five waves of pandemicPLOS ONE

Dear Dr. Mokhtari-Azad,

Thank you for submitting your manuscript to PLOS ONE. After careful consideration, we feel that it has merit but does not fully meet PLOS ONE’s publication criteria as it currently stands. Therefore, we invite you to submit a revised version of the manuscript that addresses the points raised during the review process. The language should be improved. Please submit your revised manuscript by Mar 06 2022 11:59PM. If you will need more time than this to complete your revisions, please reply to this message or contact the journal office at plosone@plos.org. Please include the following items when submitting your revised manuscript:A rebuttal letter that responds to each point raised by the academic editor and reviewer(s). You should upload this letter as a separate file labeled 'Response to Reviewers'.A marked-up copy of your manuscript that highlights changes made to the original version. You should upload this as a separate file labeled 'Revised Manuscript with Track Changes'.An unmarked version of your revised paper without tracked changes. You should upload this as a separate file labeled 'Manuscript'.

We look forward to receiving your revised manuscript.

Kind regards,

Etsuro Ito

Academic Editor

PLOS ONE

Journal Requirements:

"Authors would like to express their great thanks to all data contributors, i.e. the authors and their originating laboratories responsible for obtaining the specimens, and their submitting laboratories for generating the genetic sequence and metadata and sharing via the GISAID Initiative, on which this research is based. We thank GISAID for all their support. A part of this study is supported by NIMAD under grant number 994376."

Reviewers' comments:

Reviewer's Responses to Questions

**Comments to the Author**

1. Is the manuscript technically sound, and do the data support the conclusions?

Reviewer #1: Yes

Reviewer #2: Yes

Reviewer #3: Yes

2. Has the statistical analysis been performed appropriately and rigorously? 

Reviewer #1: N/A

Reviewer #2: Yes

Reviewer #3: N/A

3. Have the authors made all data underlying the findings in their manuscript fully available?

Reviewer #1: Yes

Reviewer #2: Yes

Reviewer #3: Yes

4. Is the manuscript presented in an intelligible fashion and written in standard English?

Reviewer #1: Yes

Reviewer #2: Yes

Reviewer #3: Yes

5. Review Comments to the Author

Reviewer #1: Regarding the manuscript entitled “Whole genome sequencing of SARS-CoV2 strains circulating in Iran during five waves of pandemic”:

The authors performed an extensive analysis of the SARS-CoV-2 genomes in Iran to find information about the SARS-CoV2 lineages, variants of interests and variants of concerns during five waves of pandemic. Overall it is a well-organized manuscript and a nice contribution to the field and the information presented in this manuscript can be beneficial to the other researchers.

Below are some suggestions that the authors may wish to consider to improve their manuscript:

-Please mention to the COVID-19 in the introduction section.

-Please mention to the different known clades of SARS-CoV-2 in the introduction section.

-In the figure legend please replace “constricted” by “constructed”

-The number of viruses in the phylogenetic tree is 53.

-Since the purpose of the manuscript was finding information about the viral lineages, variants of interests and variants of concern, it would be suggested to describe them briefly.

-Please add the following reference “Usage of peptidases by SARS-CoV-2 and several human coronaviruses as receptors: A mysterious story” in the line 366.

Reviewer #2: The manuscript presents valuable information about SARS-CoV2 viruses circulating in Iran during 5 waves of the pandemic. It is well written and organized. Only few revisions are suggested:

- The Introduction mostly explains the virus proteins and structure, while it might be preferred that first the Introduction part talks also about why this virus is important why this study was designed.

- It is suggested that the mutations in different proteins (non-structural, structural and accessory) are entered and organized in a table, so the reader can easily see and follow different mutations in different proteins during different waves of the pandemic in Iran.

- Line 486: … which the results were compatible with … -- needs to be revised to: … of which the results were compatible with …

Reviewer #3: This is a thorough analysis of the genome sequences of SARS-CoV-2 viruses in circulation in Iran over five waves of the pandemic. The full genome sequence of five to twenty viruses (10,10, 9, 5 and 20) from each wave were analysed by next generation sequencing and the results analysed to report here ion what variation has been seen.

The data quality has not been described fully: for example, how complete each genome sequence turned out to be and what proportion of undetermined nucleotides were in the genome sequence assembly. This would provide some kind of quality assessment of the gene sequence data. The gene sequence data have been shared in the EpiCov database of GISAID, but I have not checked the sequences fully.

The catalogue of changes seen and described in the results is comprehensive, but going through the results for each virus polypeptide is demanding for the reader and I wondered if the authors could come up with some kind of graphical display to supplement the description of what was seen and elaborated on in the text. I think for most virus polypeptides this could be relatively easily done, but, like in spike (for which there is a dedicated table), the variation in NSP3 seems quite extensive. Nevertheless, I think a graphical representation would enhance the message of the manuscript greatly.

In the discussion the authors summarise a lot of literature on the likely effects of amino acid substitutions. My feeling is that the authors should differentiate between results and conclusions made by experiment from those that have been generated by modelling. Moreover, there is frequent use of phrases like ‘viral oligomerisation interface’ (e.g. lines 136, 140, 141, 147, 150, 151 etc.). Here I was not clear in many cases what the evidence for this conclusion was. I might have missed the references to many of these. In addition, it was not what was meant precisely by a ‘viral oligomerisation interface’. I was not sure of this meant oligomerisation of the relevant polypeptide or something else.

There are also two places in which antibody recognition sites are described for the NSP7 and NSP8 (lines 175 and 179). Do the authors know if this recognition is by post-infection serum or something else? Also, references should be given for this. NSP7 was also stated to undergo antigenic drift. I wonder if this is better defined as antigenic change, or change in a T-cell epitope. This is also referred to on line 314.

The phylogenetic tree shown in the figure uses the GISAID clade nomenclature. I wonder if it is possible to correlate these clades and subclades with the PANGO lineages.

I think it might also be useful to include a graph of the five waves of COVID-19 in Iran and indicate when the samples were taken on the graph.

Minor points.

In many cases the authors refer to mutations in the virus polypeptides. I think, strictly speaking, mutations occur in genes that encode mutant proteins that have amino acid substitutions. I would recommend that this is considered by the authors.

There are a number of places where the English is not correct – the indefinite and the definite articles are missing in some number of places. This should be corrected by asking for additional help from colleagues.

Some other points include:

Generally, use family names only when referring to work done by others.

Also, the word ‘data’ has been used both as a singular term (incorrectly) and as a plural (correctly). There should be consistency.

I would suggest that when referring to a series of amino acid substitutions the order of these substitutions is always from the N-terminal to C-terminal of the polypeptide.

Line 62, define what a ‘primase’ is.

Line 69. I suggest saying that Spike ‘mediates’ attachment and entry rather than is ‘important for’ attachment and entry.

Line 88. I think ‘designated’ is not the right word here – perhaps ‘selected’.

Line 98. I suggest ‘Library construction was done by using…’

Line 99. Explain what ‘captured’ means in this context.

Line 236. Define in which wave hCoV-19/Iran/Gilan/NICS1-58/2020 was isolated.

Line 295. The word ‘third’ should be ‘thirds’.

Line 305. “Recommended’ seems to be the wrong word here.

Line 346. Add the words ‘change in’ ahead of ‘incubation period’.

Line 366. I think this should be Receptor Binding Domain (not Receptor Binding Motif).

Lines 455 to 456. I am not clear what intraviral connections refers to. Lines 453 to 457 might be better composed.

Line 481. The authors comment on the ‘rate’ of mutation. However, mutations rates have not been addressed in this work. This needs re-wording to reflect more accurately what has been observed in this study.

6. PLOS authors have the option to publish the peer review history of their article (what does this mean?). If published, this will include your full peer review and any attached files.

Reviewer #1: No

Reviewer #2: No

Reviewer #3: No

---

## [Author Response · Author response to Decision Letter 0]

13 Feb 2022

Dear Editor-in-Chief of PLOS ONE,

Thank you very much for the review of our manuscript entitled: “Whole genome sequencing of SARS-CoV2 strains circulating in Iran during five waves of pandemic”.

We sincerely appreciate all valuable comments and suggestions, which helped

us to improve the quality of the article. Our responses to the Reviewers’ comments 

are described below in a point-to-point manner. Appropriated changes, suggested

by the Reviewers, have been introduced to the manuscript (highlighted within the

document). The language is improved.

Dear Reviewer 1:

The changes were highlighted in yellow.

- Please mention to the COVID-19 in the introduction section.

The sentence was added to the beginning of the Introduction.

- Please mention to the different known clades of SARS-CoV-2 in the introduction section.

Was added to the last paragraph of Introduction. 

- In the figure legend please replace “constricted” by “constructed”

Done.

- The number of viruses in the phylogenetic tree is 53.

Phylogenetic tree was corrected.

- Since the purpose of the manuscript was finding information about the viral lineages, variants of interests and variants of concern, it would be suggested to describe them briefly.

Was described at the end of Introduction.

- Please add the following reference “Usage of peptidases by SARS-CoV-2 and several human coronaviruses as receptors: A mysterious story” in the line 366.

Was added.

Dear Reviewer 2:

The changes were underlined.

- The Introduction mostly explains the virus proteins and structure, while it might be preferred that first the Introduction part talks also about why this virus is important why this study was designed.

It was added to the end of Introduction.

- It is suggested that the mutations in different proteins (non-structural, structural and accessory) are entered and organized in a table, so the reader can easily see and follow different mutations in different proteins during different waves of the pandemic in Iran.

Mutations of nonstructural genes illustrated in the figure.

- Line 486: … which the results were compatible with … -- needs to be revised to: … of which the results were compatible with …

Corrected. 

Dear Reviewer 3:

The changes were highlighted in green.

This is a thorough analysis of the genome sequences of SARS-CoV-2 viruses in circulation in Iran over five waves of the pandemic. The full genome sequence of five to twenty viruses (10,10, 9, 5 and 20) from each wave were analyzed by next generation sequencing and the results analyzed to report here on what variation has been seen.

- The data quality has not been described fully: for example, how complete each genome sequence turned out to be and what proportion of undetermined nucleotides were in the genome sequence assembly. This would provide some kind of quality assessment of the gene sequence data. The gene sequence data have been shared in the EpiCov database of GISAID, but I have not checked the sequences fully.

Added to the materials and methods section.

- The catalogue of changes seen and described in the results is comprehensive, but going through the results for each virus polypeptide is demanding for the reader and I wondered if the authors could come up with some kind of graphical display to supplement the description of what was seen and elaborated on in the text. I think for most virus polypeptides this could be relatively easily done, but, like in spike (for which there is a dedicated table), the variation in NSP3 seems quite extensive. Nevertheless, I think a graphical representation would enhance the message of the manuscript greatly.

Mutations of nonstructural genes illustrated in the figure.

- In the discussion the authors summarize a lot of literature on the likely effects of amino acid substitutions. My feeling is that the authors should differentiate between results and conclusions made by experiment from those that have been generated by modelling. Moreover, there is frequent use of phrases like ‘viral oligomerisation interface’ (e.g. lines 136, 140, 141, 147, 150, 151 etc.). Here I was not clear in many cases what the evidence for this conclusion was. I might have missed the references to many of these. In addition, it was not what was meant precisely by a ‘viral oligomerisation interface’. I was not sure of this meant oligomerisation of the relevant polypeptide or something else.

That is really important point. There is not any reference about the role of some proteins as viral oligomerization interfaces, we added this, just extracted from GISAID mutation analysis, then we deleted them.

- There are also two places in which antibody recognition sites are described for the NSP7 and NSP8 (lines 175 and 179). Do the authors know if this recognition is by post-infection serum or something else? Also, references should be given for this. NSP7 was also stated to undergo antigenic drift. I wonder if this is better defined as antigenic change, or change in a T-cell epitope. This is also referred to on line 314.

It is another important point which is written just according to GISAID mutation analysis and there is not publication or in vitro studies, then we removed those from the article.

- The phylogenetic tree shown in the figure uses the GISAID clade nomenclature. I wonder if it is possible to correlate these clades and subclades with the PANGO lineages.

 We could change the clades to PANGO in the tree, but because we used mostly 

 clades in the text, then we prefer to put clades, if it is necessary we could

 change them.

- I think it might also be useful to include a graph of the five waves of COVID-19 in Iran and indicate when the samples were taken on the graph.

It is useful to include a graph, but because of the article length and number of tables and figures, we just wrote the date of each wave and the number of assessed samples in each wave. 

Minor points.

- In many cases the authors refer to mutations in the virus polypeptides. I think, strictly speaking, mutations occur in genes that encode mutant proteins that have amino acid substitutions. I would recommend that this is considered by the authors.

 All were corrected.

- There are a number of places where the English is not correct – the indefinite and the definite articles are missing in some number of places. This should be corrected by asking for additional help from colleagues.

Reviewed and corrected as much as possible.

Some other points include:

- Generally, use family names only when referring to work done by others.

All were corrected.

- Also, the word ‘data’ has been used both as a singular term (incorrectly) and as a plural (correctly). There should be consistency.

Corrected.

I would suggest that when referring to a series of amino acid substitutions the order of these substitutions is always from the N-terminal to C-terminal of the polypeptide.

 Corrected

- Line 62, define what a ‘primase’ is.

Added.

- Line 69. I suggest saying that Spike ‘mediates’ attachment and entry rather than is ‘important for’ attachment and entry.

Corrected.

- Line 88. I think ‘designated’ is not the right word here – perhaps ‘selected’.

Corrected.

- Line 98. I suggest ‘Library construction was done by using…’

Corrected.

- Line 99. Explain what ‘captured’ means in this context.

The sentence was changed to clear explanation.

- Line 236. Define in which wave hCoV-19/Iran/Gilan/NICS1-58/2020 was isolated.

Cleared.

- Line 295. The word ‘third’ should be ‘thirds’.

Corrected. 

- Line 305. “Recommended’ seems to be the wrong word here.

Corrected.

- Line 346. Add the words ‘change in’ ahead of ‘incubation period’.

Added.

- Line 366. I think this should be Receptor Binding Domain (not Receptor Binding Motif).

Corrected.

- Lines 455 to 456. I am not clear what intraviral connections refers to. Lines 453 to 457 might be better composed.

Corrected.

- Line 481. The authors comment on the ‘rate’ of mutation. However, mutations rates have not been addressed in this work. This needs re-wording to reflect more accurately what has been observed in this study.

 It corrected. We mean the number of mutations.

---

## [Decision Letter · Decision Letter 1]

18 Mar 2022

PONE-D-21-38222R1Whole genome sequencing of SARS-CoV2 strains circulating in Iran during five waves of pandemicPLOS ONE

Dear Dr. Mokhtari-Azad,

Thank you for submitting your manuscript to PLOS ONE. After careful consideration, we feel that it has merit but does not fully meet PLOS ONE’s publication criteria as it currently stands. Therefore, we invite you to submit a revised version of the manuscript that addresses the points raised during the review process. Please fined attached.This file show the suggestions by one of the reviewers.

We look forward to receiving your revised manuscript.

Kind regards,

Etsuro Ito

Academic Editor

PLOS ONE

Journal Requirements:

Reviewers' comments:

Reviewer's Responses to Questions

**Comments to the Author**

1. If the authors have adequately addressed your comments raised in a previous round of review and you feel that this manuscript is now acceptable for publication, you may indicate that here to bypass the “Comments to the Author” section, enter your conflict of interest statement in the “Confidential to Editor” section, and submit your "Accept" recommendation.

Reviewer #1: All comments have been addressed

Reviewer #3: All comments have been addressed

2. Is the manuscript technically sound, and do the data support the conclusions?

Reviewer #1: Yes

Reviewer #3: Yes

3. Has the statistical analysis been performed appropriately and rigorously? 

Reviewer #1: Yes

Reviewer #3: Yes

4. Have the authors made all data underlying the findings in their manuscript fully available?

Reviewer #1: Yes

Reviewer #3: Yes

5. Is the manuscript presented in an intelligible fashion and written in standard English?

Reviewer #1: Yes

Reviewer #3: No

6. Review Comments to the Author

Reviewer #1: (No Response)

Reviewer #3: The manuscript by Yavarian et al. has addressed many of the points I raised in my report. However, I think that there are still various sections of the manuscript where improvements to the flow or meaning can be made. I will attach a scan of a handwritten marked-up document for the authors to consider. However, there are places that the authors still need to address.

The main places where the authors need to further modify the text are as follows. (Minor changes suggested are in the marked-up copy).

Lines 108 to 110. The term ‘mutations’ remain in this section to be used for amino acid substitutions. The same is also seen in lines 113 and 116. I suggest the authors re-check for the appropriate usage. I have made some suggestions in the marked-up scanned copy.

Line 140. Introduce the concept here of amino acid deletions e.g. using the phrase ‘a deletion of G66 (G66del) and S76 (S67del)…’ See also lines 162 to 164, and line 240

Lines 144 and 145. Introduce the concept of stop codons in NS7b and NS8 - and the effect of the stop codons on what can be expected to be translated.

Line 171 is not clear what the authors are trying to say here. Perhaps it is that amino acid substitutions in NS3 deserve greater study in vitro.

Lines 195 to 197. I have made suggestions for changes here in the marked copy.

Line 211. It was not clear what the authors meant by ‘missense mutations’.

Line 243. A reference is needed.

Line 271. It was unclear what therapeutic effects were envisaged here.

Line 276. Here the authors are presumably referring to amino acid substitutions in the nucleoprotein, not in the gene.

Line 277. A reference is needed.

Line 287. A reference is needed.

Lines 289 to 291. It is not clear what the authors are saying that termination of the NS8 gene somehow affects spike affinity for the receptor. The authors should elaborate on this.

7. PLOS authors have the option to publish the peer review history of their article (what does this mean?). If published, this will include your full peer review and any attached files.

Reviewer #1: No

Reviewer #3: No

---

## [Author Response · Author response to Decision Letter 1]

13 Apr 2022

Dear Dear Editor-in-Chief of PLOS ONE,

Thank you very much for the review of our manuscript entitled: “Whole genome 

sequencing of SARS-CoV2 strains circulating in Iran during five waves of 

pandemic”.

We sincerely appreciate all valuable comments and suggestions, especially dear reviewer 3, which helped us to improve the quality of the article. Our responses to the Reviewers’ comments are described below in a point-to-point manner. Appropriated changes, suggested by the Reviewers, have been introduced to the manuscript. The language is improved by all corrections made by reviewer 3. 

Reviewer #3: The manuscript by Yavarian et al. has addressed many of the points I raised in my report. However, I think that there are still various sections of the manuscript where improvements to the flow or meaning can be made. I will attach a scan of a handwritten marked-up document for the authors to consider. However, there are places that the authors still need to address.

The main places where the authors need to further modify the text are as follows. (Minor changes suggested are in the marked-up copy).

Lines 108 to 110. The term ‘mutations’ remain in this section to be used for amino acid substitutions. The same is also seen in lines 113 and 116. I suggest the authors re-check for the appropriate usage. I have made some suggestions in the marked-up scanned copy.

They all corrected.

Line 140. Introduce the concept here of amino acid deletions e.g. using the phrase ‘a deletion of G66 (G66del) and S76 (S67del)…’ See also lines 162 to 164, and line 240.

They all corrected.

Lines 144 and 145. Introduce the concept of stop codons in NS7b and NS8 - and the effect of the stop codons on what can be expected to be translated.

We added that stop codons lead to truncated proteins.

Line 171 is not clear what the authors are trying to say here. Perhaps it is that amino acid substitutions in NS3 deserve greater study in vitro.

It was re-written.

Lines 195 to 197. I have made suggestions for changes here in the marked copy.

Correction was made.

Line 211. It was not clear what the authors meant by ‘missense mutations’.

We wrote change in amino acid.

Line 243. A reference is needed.

Added.

Line 271. It was unclear what therapeutic effects were envisaged here.

It was not appropriate sentence, we removed it.

Line 276. Here the authors are presumably referring to amino acid substitutions in the nucleoprotein, not in the gene.

Corrected.

Line 277. A reference is needed.

Added.

Line 287. A reference is needed.

Added. 

Lines 289 to 291. It is not clear what the authors are saying that termination of the NS8 gene somehow affects spike affinity for the receptor. The authors should elaborate on this.

We re-wrote it.

---

## [Editor Report · Decision Letter 2]

18 Apr 2022

Whole genome sequencing of SARS-CoV2 strains circulating in Iran during five waves of pandemic

PONE-D-21-38222R2

Dear Dr. Mokhtari-Azad,

We’re pleased to inform you that your manuscript has been judged scientifically suitable for publication and will be formally accepted for publication once it meets all outstanding technical requirements.

Kind regards,

Etsuro Ito

Academic Editor

PLOS ONE

---

## [Editor Report · Acceptance letter]

22 Apr 2022

PONE-D-21-38222R2 

Whole genome sequencing of SARS-CoV2 strains circulating in Iran during five waves of pandemic 

Dear Dr. Mokhtari-Azad:

I'm pleased to inform you that your manuscript has been deemed suitable for publication in PLOS ONE. Congratulations! Your manuscript is now with our production department. 

Kind regards, 

on behalf of

Prof. Etsuro Ito 

Academic Editor

PLOS ONE